# Does the Tourism Development of a Destination Determine Its Socioeconomic Development? An Analysis through Structural Equation Modeling in Medium-Sized Cities of Andalusia, Spain

**Juan Antonio Parrilla-González**

Department of Economy, University of Jaén, 23071 Jaén, Spain; japarril@ujaen.es

**Abstract:** Medium-sized European cities have been playing an increasingly significant role in the economic development of countries in recent decades, establishing themselves as genuinely specialized local production systems with great potential for stimulating the economy and generating added value. In many of these cities, in addition, tourism has become an incredibly strong economic activity with the capacity to stimulate local economies, as it contributes to the enhancement of endogenous resources and the generation of a multiplier effect on other economic sectors. This paper uses a structural equation model to demonstrate, first, that a direct relationship exists between tourism development and economic development and second, that, of all cities analyzed (medium-sized cities of Andalusia, Spain), those with a higher level of tourism development are actually those showing a higher level of socioeconomic development, which confirms that tourism has great potential as a tool for endogenous development.

**Keywords:** medium-sized cities; socioeconomic development; tourism development; tourism destination; structural equation modelling; Andalusia

## 1. Introduction

One of the major changes that have occurred within the framework of the theory of economic development over the last twenty years is the consolidation of a new paradigm known as territorial, endogenous or local development [1]. According to this paradigm, far from eliminating differences between areas, the process of globalization is stimulating the expansion of all of its forms, consistent with the new spatial logic of global capitalism [2–4]. Some authors highlight the great growth of medium-sized cities and their influence on urban and socioeconomic development [5–7].

According to the national urban networks in Europe, the European Commission [8], highlights small and medium-sized European cities as centers for the development of industrial activities and services for research and technology, and for tourism and leisure. In recent decades, these cities have reaffirmed their role. Thus, they function as regional centers that must cooperate as part of a polycentric model, in order to ensure their added value compared to other cities located in rural and peripheral areas, as well as in areas with specific geographical challenges and needs [9,10].

In this sense, now, and in line with what happened in other parts of Europe, Andalusia (Spain) is embarking on a process of structural change in which intermediate cities are becoming increasingly visible, which is based, inter alia, on the enhancement of the endogenous resources serving tourism development [11], with the conviction that this activity has a strong dynamic effect on the economy as a whole.

In this way, tourism has favored the proliferation of research on how this process contributes to the social, economic and cultural well-being of the inhabitants of these cultural heritage sites, focusing mainly on large monumental cities although, more recently, on medium-sized cities [12].

The capacity of tourism as a lever for development in medium-sized cities has made one of the most important consequences of the progressive incorporation of tourism in the whole of the Spanish Mediterranean coast. This has led to the generation of production specialization processes in many of these cities [13].

In fact, the whole of the Spanish Mediterranean coast has become, after 60 years of continuous development, one of the densest regions in Europe. Some authors speak of a long and compact "linear city", made up of a conglomerate of hotels, restaurants and leisure facilities based almost exclusively on tourism [14].

Tourism, understood as an economic activity, has spread throughout the settlement system and has contributed directly to the strengthening of the network of medium-sized cities, which in turn drives the phenomenon of urban deconcentration on a regional scale. These medium-sized cities that have been organized by capturing seasonal (tourist) flows have given rise to places with favorable conditions for the residential location of certain segments of the population [15]. In this way, the set of tourist municipalities that, in the mid-twentieth century, constituted towns with temporary visitors, have ended up being medium-sized cities or places with consolidated urban attributes [13,16].

Thus, tourism is a dynamic tool of the territory for this region, so that for medium cities, it can represent an important advance in terms of socio-economic development, especially in the case of medium cities that are the object of this study. In this context, and taking as reference Pulido and Parrilla studies [17], the object of the research is the relationship between tourism development and socioeconomic development, focusing on medium-sized cities.

The hypothesis of this research is that the level of tourism development of an area (in this particular research study, the medium-sized cities of Andalusia) affects its level of socioeconomic development. Put another way, those territories with a higher level of tourism development are also those showing a higher level of socioeconomic development, which would demonstrate that tourism is an important instrument of endogenous development.

To test this hypothesis, several indicators aimed at measuring the level of tourism development and socioeconomic development of the medium-sized cities of Andalusia will be developed. Then, it will be verified whether any relationship exists between both indicators, and these cities will be classified in order to draw some conclusions.

## 2. Literature Review and Theoretical Implications

### 2.1. Medium-Sized Cities as a Potential Territory of Urban Development

From the territorial point of view, in Europe, which is undergoing a process of enlargement of the European Union, a decentralization process has been taking place within many states in favor of regions and cities, resulting in an increase in their power and relevance, which provides them with the necessary resources and skills to generate responses through their management model [18]. Until relatively recently, the city was studied as part of the landscape, considering, mainly, its descriptive and morphological aspects, and ignoring the interpretative ones.

It was in the seventies that there was an unprecedented expansion of low-density suburban areas, which, in a "macro-level" analysis, may seem diffuse and networked. However, when observed at "micro"-level, each node of that network shows specific characters and different organizational models [19]. In this context, medium and small-sized cities have the opportunity to present themselves as effective, specialized local production systems [20,21], bearing in mind that many of these cities lack adequate strategic resources, which limits their innovativeness [22].

It is important to mention the European Territorial Strategy [23] as a starting point when building a European urban system in which non-metropolitan cities, known as intermediate cities, take on special significance. Talking about intermediate cities means dealing with a crucial step in the road that society has slowly followed to get to the current organization of the space, customs, ways of thinking and doing, and of the current scale

of values, which highlights their potential, both because of the population size and their ability to serve as an intermediary between the metropolis and the rural world [24].

The development of small and medium-sized cities is an important aspect of healthy urban agglomeration in metropolitan areas [25]. To improve the economic growth of small and medium-sized cities and the differences in the levels of economic development, the study of national urban networks is essential [26]. This urban development of medium-sized cities in Andalusia has been possible due to their being in areas of potential development for the tourism industry [16].

## 2.2. Tourism as an Endogenous Development Tool

In a context characterized by the great paradigm of the globalization of the economy and society, the need for bringing a global dimension to local territories and markets should be considered as an essential feature of economic and social integration, competition between economies and innovation among territories.

For Vazquez-Barquero and Rodriguez-Cohard [27], the process of globalization means increased competition within territories, given that companies do not compete in isolation, but rather in conjunction with the productive and institutional environment to which they belong. These authors emphasize the process of local economic development as a process of growth and structural change that occurs as a result of the transfer of resources from traditional to modern activities, the use of external economies and the introduction of innovations, which generates an increase in the level of welfare of the population of a city or region [28].

Moreover, in recent years, tourism development is becoming a means for revitalizing the territory, as it has a number of features that make it possible to generate new opportunities at the local and regional level [29], as well as at the global level [30]. The forces of globalization have contributed to the restructuring of rural economies and peripheral regions of the developed world, which are, thus, in line with the forces that have contributed to the growth of international tourism [31].

The rapid growth of the local tourism industry has been perceived by many rural communities as an opportunity to develop new economic activities, as this growing and feasible tourism development is consistent with their resources and their needs [32,33].

The potential to expand business opportunities in tourism plays a key role in developing a territory from a tourism perspective, where this characteristic is perceived by the communities and regions. Nevertheless, in most cases, this perception fails to be implemented and, therefore, it is not perceived as a real opportunity for local development. What is more, many of the arguments included in the concept of tourism development have to do with local development alternatives compared to other economic activities [34].

In short, tourism development is gaining importance in recent years, due to a variety of factors, including the rapid growth of the tourism industry in rural areas, the availability of tourism resources that shape the tourism destination, and the activity of the different businesses within the territory.

## 2.3. Tourism Development as a Socioeconomic Development Tool

The third report on territorial development in Andalusia [35] recognizes that tourism plays an increasingly relevant role in the regional development of this region, where the dynamism of tourism has become a crucial factor in explaining and understanding many of the territorial, economic, social and cultural processes that have taken place in recent years.

This justifies the need to analyze the relationship between tourism development and socioeconomic development on the basis of the conceptual and analytical support offered by the idea of a tourism destination. Pulido-Fernández [36] highlights the evolution experienced by tourism destinations from the scientific perspective, given that, until recently, all research studies focused on their microeconomic dimension, or macroeconomic aggregates, relegating the role of destinations (which have mesoeconomic perspectives) as key elements in the development of tourism to a peripheral role.

Thus, it has been accepted that tourism destinations are territorial systems organized through a complex network of actors that provide a set of goods and services that are able to satisfy the complex needs of the tourist. However, a destination is not just a territory where there is a group of stakeholders, more or less coordinated and organized, involved in the production and supply of tourism products; instead, it is actually perceived in the collective imagination of potential customers as the territory in which to enjoy a memorable experience.

Therefore, and as a conclusion, the tourism destination is the productive area that has to be taken into account when analyzing the relationships between tourism development and socioeconomic development, since this is where the tourism event is to be found for both production and consumption, which justifies the importance of tourism as a socioeconomic development tool [37].

## 3. Materials and Methods

To carry out the appropriate methodology in this study, the research by Pulido and Parrilla [17] has been taken as a reference, with the particular difference that in our case, the study is focused on medium-sized cities. The concept of a medium-sized city depends on the territorial framework and the aspects taken into account for its conceptualization. In the case of Andalusia, and from the economic and geographic perspective given [20], the following characteristics are considered:

− Population size
− Population growth in recent years
− Capacity for territorial planning in relation to the urban functions performed
− Economic potential, degree of industrialization and specialization

Given these parameters, and acknowledging the importance of the configuration of these types of urban structures in Andalusia, medium-sized cities are those with populations between 10,000 and 90,000 inhabitants, that show a rapid rate of population growth, that are sometimes located near large metropolitan areas whose capacities for territorial planning have been established not only on the basis of the role of each population center within the system of cities, but also of the equipment operating as intermediate centers with the capacity to organize the environment.

Another important feature is the consideration of newly established companies, the jobs they create and the number of exporting firms, establishing the economic potential of medium-sized cities to determine their economic dynamism linked to the territory.

On the basis of the above, Table 1 shows the medium-sized cities analyzed in this paper, divided within the provinces of Andalusia.

Thus, first of all, the levels of tourism development and socioeconomic development of the selected cities have been analyzed. Then, it has been determined whether any relationship exists between both indicators. Finally, a classification of these cities according to the type of relationship that exists between these two latent variables has been presented, which allows for conclusions to be drawn that validate our initial hypothesis.

**Table 1.** Medium-sized cities under study (cities per province).

| Provincia | Ciudades Medias |
|---|---|
| Almería | Adra, Albox, Berja, Cuevas de Almanzora, El Ejido, Huércal de Almería, Huercal-Overa, Níjar, Roquetas de Mar, Vera, Vícar. |
| Cádiz | Arcos de la Frontera, Barbate, Chiclana de la Frontera, Chipiona, Conil de la Frontera, El Puerto de Santa María, Jimena de la Frontera, La Linea de la Concepción, Los Barrios, Medina-Sidonia, Puerto Real, Rota, San Fernando, San Roque, San Lucar de Barrameda, Tarifa, Ubrique, Véjer de la Frontera, Villamartín. |
| Córdoba | Aguilar de la Frontera, Baena, Cabra, Fuente Palmera, La Carlota, Lucena, Montilla, Palma del Rio, Peñarroya-Pueblonuevo, Pozoblanco, Priego de Córdoba, Puente Genil, Rute. |
| Granada | Albolote, Almuñecar, Armilla, Atarfe, Baza, Churriana de la Vega, Guadix, Huétor Tajar, Huétor Vega, Íllora, La Zubia, Las Gabias, Loja, Maracena, Motril, Ogíjares, Peligros, Pinos Puente, Salobreña, Santa Fé. |
| Jaén | Alcalá la Real, Alcaudete, Andújar, Baeza, Bailén, Jódar, La Carolina, Linares, Mancha Real, Martos, Torredelcampo, Torredonjimeno, Úbeda, Villacarrillo. |
| Málaga | Alhaurín de la Torre, Alhaurín el Grande, Álora, Antequera, Benalmádena, Cártama, Coín, Estepona, Fuengirola, Manilva, Mijas, Nerja, Rincón de la Victoria, Ronda, Torremolinos, Torrox, Velez-Málaga. |
| Sevilla | Alcalá de Guadaira, Alcalá del Río, Arahal, Bormujos, Brenes, Camas, Cantillana, Carmona, Castilleja de la Cuesta, Coria del Río, Écija, El Viso del Alcor, Espartinas, Estepa, Gines, La Algaba, La Puebla de Cazalla, La Puebla del Río, La Rinconada, Las Cabezas de San Juan, Lebrija, Lora del Río, Los Palacios y Villafranca, Mairena del Alcor, Marchena, Morón de la Frontera, Osuna, Pilas, San Juan de Aznalfarache, Sanlúcar la Mayor, Tomares, Utrera. |
| Huelva | Aljaraque, Almonte, Ayamonte, Bollullos, Cartaya, Gibraleón, Guillena, Isla Cristina, La Palma del Condado, Lepe, Moguer, Punta Umbría, Valverde del Camino. |

Source: author's own elaboration.

### 3.1. Selection of Indicators

An empirical work seeking to determine whether the level of tourism development of a territory (in our research, medium-sized cities of Andalusia) determines its level of economic development should be performed using a sufficiently long-time horizon that allows meaningful measurement of the influence of the variables used in this research (in 15 years). For this reason, in this study, the time horizon comprises the period from 2004 to 2019. The reason for choosing this time horizon is the availability of the indicator and the comparison of its evolution in this period of time.

In the present study, two latent variables are considered, which are called tourism development and socioeconomic development. These variables are determined by sixty-two manifest variables. Specifically, the tourism development variable is expressed in terms of thirty-nine of these manifest variables, while the socioeconomic development variable has been measured by the remaining thirty-three indicators.

These indicators have been chosen, taking into account the limitations that exist with regard to the availability of local information. The two main statistical sources providing information at the municipal level in Andalusia (National Institute of Statistics and Institute of Statistics and Cartography of Andalusia) have been consulted. These sources have redirected the selection of indicators of tourism development and socioeconomic development to other primary sources which provide individualized information on each area of study (tourism, economy, innovation, society, social welfare, environment), which enable, in general, an approximation of the measurement of the latent variables under study.

The full list of indicators (and their corresponding sources) that has been considered for each of the two latent variables can be found in Table 2.

**Table 2.** Variables used to measure tourism development and socioeconomic development.

| Tourism Development | Source |
| --- | --- |
| Museums | Ministry of Education, Culture and Sports |
| Hotel Rooms | Regional Ministry of Tourism and Commerce |
| Hotel-Apartment Rooms | Regional Ministry of Tourism and Commerce |
| B&B and Guest House Rooms | Regional Ministry of Tourism and Commerce |
| 4-key Apartment Rooms | Regional Ministry of Tourism and Commerce |
| 3-key apartment Rooms | Regional Ministry of Tourism and Commerce |
| 2-key apartment Rooms | Regional Ministry of Tourism and Commerce |
| 1-key apartment Rooms | Regional Ministry of Tourism and Commerce |
| Unemployment in Tourism | ARGOS Observatory-Regional Ministry of Economy, Innovation, Science and Employment |
| Cinemas | AIMC. Media Research Association |
| Film Screens | AIMC. Media Research Association |
| Seating Capacity of Cinemas | AIMC. Media Research Association |
| Banks | Bank of Spain. Statistical Bulletin |
| Savings Bank | Bank of Spain. Statistical Bulletin |
| Credit Unions | Bank of Spain. Statistical Bulletin |
| Campsites | Regional Ministry of Tourism and Commerce |
| Hotels | Regional Ministry of Tourism and Commerce |
| Hotel-Apartment | Regional Ministry of Tourism and Commerce |
| B&B and Guest Houses | Regional Ministry of Tourism and Commerce |
| Restaurants | Regional Ministry of Tourism and Commerce |
| Cafés | Regional Ministry of Tourism and Commerce |
| Taxis | Regional Ministry of Public Works and Housing |
| Car and Driver Rental | Regional Ministry of Public Works and Housing |
| Ambulances | Regional Ministry of Health |
| Public Transp. + 10 travelers | Regional Ministry of Public Works and Housing |
| Public Transp. − 10 travelers | Regional Ministry of Public Works and Housing |
| Tax on Business Activities, Division 6 | Regional Ministry of Economy, Innovation, Science and Employment |
| Tourism Index | Spanish Economic Yearbook—La Caixa |
| Tourist offices | Regional Ministry of Tourism and Commerce |
| **Socioeconomic Development** | **Source** |
| Population Census | National Institute of Statistics |
| Municipal Register of Inhabitants Natural Population Growth | National Institute of Statistics |
| Annual Personal Income Tax | Andalusian Institute of Statistics and Cartography |
| Registered Unemployment | Tax Agency |
| Electricity Consumption | ARGOS Observatory-Regional Ministry of Economy, Innovation, Science and Employment |
| Private Cars | Sevillana—Endesa |
| Motorcycles | Directorate General for Traffic |
| Vans and Lorries | Directorate General for Traffic |
| Buses | Directorate General for Traffic |
| Commercial Vehicles | Directorate General for Traffic |
| Other Vehicles | Directorate General for Traffic |
| Buildings | Directorate General for Traffic |
| Real Estate | National Institute of Statistics—Census of Population |
| Tax on Business Activities | National Institute of Statistics—Census of Population |
| Patents | Regional Ministry of Economy, Innovation and Science |
| Utility Models | Spanish Patent and Trademark Office |
| CNAE Establishments | Spanish Patent and Trademark Office |
| Economic Activity Index | Andalusian Institute of Statistics and Cartography |
| Population Older than 65 | Economic Yearbook of Spain—La Caixa |
| Foreign Population | National Institute of Statistics |
| Marriages | National Institute of Statistics |
| Health Centers | Andalusian Institute of Statistics and Cartography |
| Vaccination Sites | Regional Ministry of Health |
| Local Clinics | Regional Ministry of Health |

| Tourism Development | Source |
|---|---|
| Public Education Institutions | Regional Ministry of Health |
| Public Libraries | Regional Ministry of Education, Culture and Sports |
| Pharmacies | Regional Ministry of Education, Culture and Sports |
| Peripherals Specialty Centers | Regional Ministry of Health |
| Public Hospitals | Regional Ministry of Health |
| Private Hospitals | Regional Ministry of Health |
| Public Hospital Beds | Regional Ministry of Health |
| Private Hospital Beds | Regional Ministry of Health |
| - | Regional Ministry of Health |

Source: Author own elaboration.

Having selected the indicators, we calculated the relative rate of change of each one of them for the period 2004–2019. There is a total of $n = 140$ observations corresponding to an equal number of medium-sized Andalusian cities. For each locality, $p + q = 62$ variables of tourism development and socioeconomic development have been measured in two time periods, that is, tinitial and tfinal; their corresponding relative rate of change has been calculated, according to the following expression:

$$RRC = \frac{X^i_{t_{final}} - X^i_{t_{initial}}}{X^i_{t_{initial}}}, \quad i = 1, \ldots, 62$$

Needless to say, all the observed features are quantitative in nature, so their relative rates of change are also quantitative variables. Specifically, in contrast to most of the features initially observed, rates are continuous quantitative variables, and their range of variation is the entire real space. These rates are, besides, dimensionless and are expressed as decimal values.

Finally, it should be noted that it has been taken into consideration the positive or negative sign that applies to the direct or inverse relationship of each indicator with the two latent variables analyzed (tourism development and socioeconomic development).

### 3.2. Structural Equation Modeling

Structural equation models (hereinafter referred to as SEM) [38–41] allow researchers to measure the relationships that occur between a set of independent variables and a set of dependent variables, as well as to determine the level of support that a sample of observations provides to the hypothesis of causality between latent variables. These models are used as confirmatory tools aimed at checking the different dependency relationships existing between the variables, in this case, tourism development and socioeconomic development.

Given that the overall aim is to check the level of support that the sample of observations provides to the hypothesis of causality between tourism development and socioeconomic development, tourism development is considered as an exogenous variable and it will be denoted by $\xi_1$, while socioeconomic development will play the role of an endogenous variable and will be denoted by $\eta_1$. It is possible to make a model of this situation by means of a diagram of paths or trajectories, as shown in Figure 1.

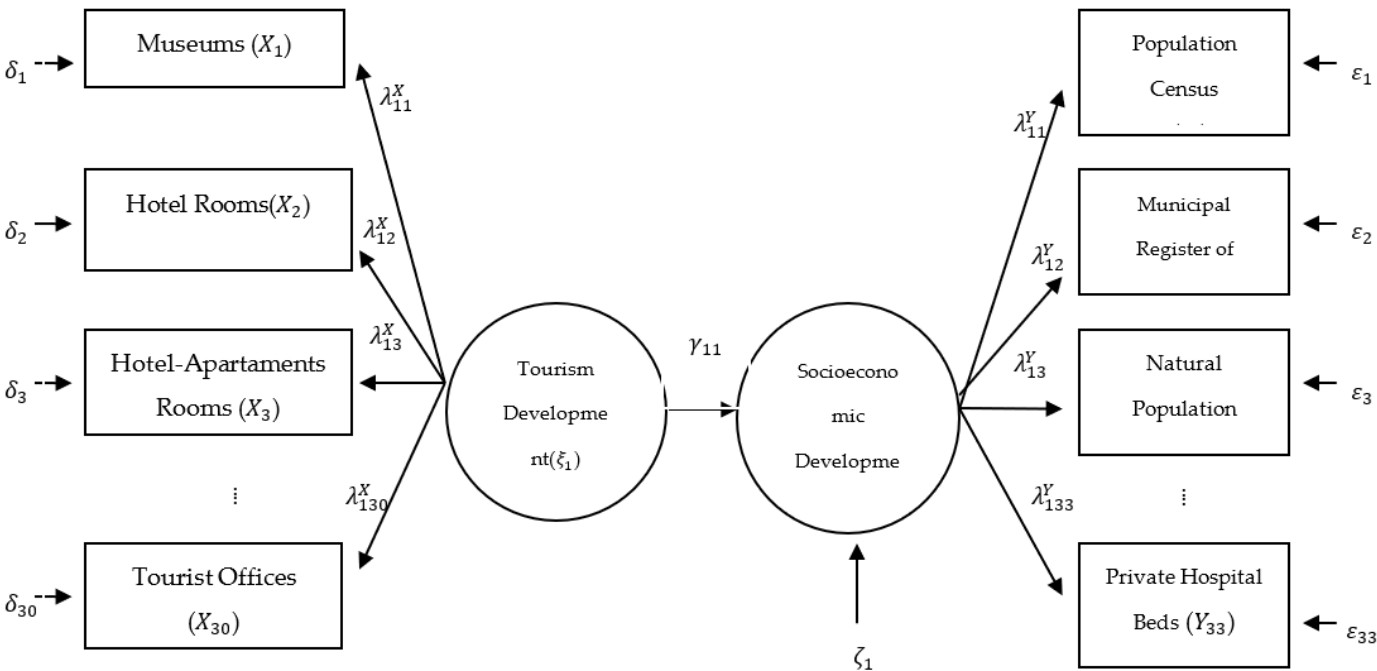

**Figure 1.** Path diagram. Source: author's own elaboration.

### 3.3. Factor Analysis of Principal Components

In the principal component analysis (hereinafter referred to as PCA), the primary objective is to maximize the variance of a linear combination of variables. Suppose we have a sample of *n* observation vectors $y_1 = (y_{11}, \ldots, y_{1p})'$, \ldots, $y_n = (y_{n1}, \ldots, y_{np})'$, so that each observed vector is a point cloud in a *p*-dimensional space. Assuming *y* as having an ellipsoidal distribution (only for better geometric visualization, as the PCA may be applied with any distribution of *y*), if the variables $y_1, \ldots, y_p$, for each vector $y\_i$ are correlated, the ellipsoidal point cloud is not oriented parallel to any of the axes represented by the variables. So, we try to find the natural axes of the point cloud whose origin coincides with the centroid of the ellipsoid, $\bar{y}$, that is, the axes of the ellipsoid. This can be done through the translation of the origin to $\bar{y}$, and later, through the rotation of the axes. After this rotation, in which the new axes become the natural axes of the ellipsoid, the new variables, that is, the main components, are uncorrelated, which means that the principal components' variance–covariance matrix is diagonal.

The rotation of the axes can be performed by multiplying the variables by an orthogonal matrix A:

$$z_i = Ay_i$$

so that the distance from the origin is invariant.

It can be seen that the orthogonal matrix that transforms $y_i$ into $z_i$ is none other than the transpose of the matrix whose columns are the normalized eigenvectors of the variance–covariance matrix of the original dataset. When the variables present significantly different variances, or when the measurement units vary, the eigenvectors are extracted from the correlation matrix to obtain a more balanced representation.

Thus, it is possible to calculate as many principal components as measured variables, so that the first of these principal components explains the greater proportion of variance of all principal components; the second principal component explains the greater proportion of variance that the first component has not been able to explain, and so on. Generally, based on the assumption that the variables are highly correlated among them, the proportion of variance explained by the last principal components will be very small, so it will be possible to discard some of them and represent the sample data using less than *p* dimensions.

*3.4. Statistical Software*

In order to carry out this analysis, version 3.0.1 of the free statistical software R has been used. This is a modular program providing basic functionality that can be extended by downloading and installing a variety of additional packages that allow performing many statistical analyses. Within this context, a package can be defined as a group of functions that together solve a common problem. Among these packages, we find lavaan, which enables fitting different models involving latent variables, such as confirmatory factor analysis or structural equation modeling, among others. This has been, therefore, the package used for the data analysis.

The analysis has been carried out in two parts. In the first, a structural equation modeling analysis (SEM) is carried out to measure the possible relationship between tourism development and socioeconomic development, and later a factor analysis of the main components is carried out with the aim of obtaining a ranking of medium-sized cities based on tourism development and socioeconomic development to obtain greater results and, therefore, a broader discussion and conclusions.

## 4. Results and Discussion

After a descriptive analysis of the data, it is observed that most of the variable means fluctuate around zero. This is due to the fact that most of the values of the RRC (relative rate of change) vary between $-1$ and 1. It is possible to identify a group of variables (Population census, Natural Population Growth, Foreign Population, Annual Personal Income Tax, Commercial Vehicles, Hotel Rooms, Apartment-Hotel Rooms, 1-Key Apartment rooms, Film Screens) showing an unusually high variance. This is explained by the existence of extreme values far from the bulk of the observations, which implies a very different evolution of the medium-sized cities analyzed as far as these variables are concerned. After checking the data and ensuring that these extreme observations are not the result of any kind of error, but are actual values of the variables, the implementation of the structural equation analysis itself is carried out.

We began with the formulation of the model. In this case, the structural model follows the equation that is shown below:

$$\eta_1 = \gamma_{11}\xi_1 + \zeta_1$$

Meanwhile, the measurement model is given by the following equations:

$$X_1 = \lambda_{11}^X \, \xi_1 + \delta_1$$

$$X_2 = \lambda_{12}^X \, \xi_1 + \delta_2$$

$$X_{29} = \lambda_{129}^X \, \xi_1 + \delta_{29}$$

$$Y_1 = \lambda_{11}^Y \, \eta_1 + \varepsilon_1$$

$$Y_2 = \lambda_{12}^Y \, \eta_1 + \varepsilon_2$$

$$Y_{33} = \lambda_{133}^Y \, \eta_1 + \varepsilon_{33}$$

It remains now to be seen whether it is possible to identify the model. To do this, the necessary condition for the identification will be verified. While it is true that these conditions do not guarantee the identification of the model in all cases (they are only necessary, and not necessary and sufficient), it has been experimentally proven that the vast majority of models that meet these conditions happen to be identifiable.

The most important of these conditions states that the number of parameters to be estimated has to be less than or equal to the number of non-redundant elements of the

sample variance–covariance matrix. In this case, the estimation of a total of 125 parameters, distributed as follows, is required:

$$29\left(\lambda^X\right) + 29(\delta) + 33\left(\lambda^Y\right) + 33(\varepsilon) + 1(\gamma) = 125$$

while the variance–covariance matrix includes a total of 1953 non-redundant elements, given that:

$$\frac{(p+q) * (p+q+1)}{2} = \frac{(33+29) * (33+29+1)}{2} = \frac{3096}{2} = 1953$$

Therefore, the first condition is fulfilled.

Another important condition is that relating to the number of indicators per latent variable. It is recommended that a minimum of three indicators per latent variable be used, and that each one of the indicators load only on one latent variable.

Moreover, regarding the metrics of the latent variables, their variances have been set to one, not only to satisfy the identification condition, but also to favor the convergence of the method of the parameter estimation. Finally, it is considered that the parameters of the regression coefficients of the indicators on their respective error terms are all equal to one.

Given the compliance of the model with the above four conditions, the probability that it can be identified is very high. We will check whether it is actually possible to estimate all the parameters that make up the model.

The tables presented below include the estimates of the parameters calculated using the maximum-likelihood method. For all parameters, their estimations and their standard errors, which have been calculated using a bootstrap or resampling method, are shown.

The *p*-value associated with the statistic Z, which contrasts the significance of the parameter, is also shown for parameters $\lambda^X$ (Table 3), $\lambda^Y$ (Table 4) and $\gamma$ (Table 5). According to Table 5, parameter $\gamma$ indicates that the sample of observations would support the hypothesis of causality between tourism development and economic development, since the value of $\gamma_{11}$ is significantly different from zero.

The results obtained should, however, be treated with caution, given that the high number of indicators loaded on each of the latent variables could be masking the true relations between them. As Hoyle [42] points out, there seems to be agreement among researchers regarding the consideration of a minimum of three indicators per latent variable for the structural equation analysis to be carried out without problems. There is no consensus, however, on whether a maximum number of indicators per factor exist. Yet, between five and ten manifest variables are usually considered for each latent variable. In this case, this number is significantly higher, so a re-specification of the model would be advisable.

As it can be seen in the previously shown tables, only eight of the tourism development indicators are associated with a parameter that is significant at a 95% confidence level. Something similar happens with socioeconomic development, for which only sixteen out of the thirty-three indicators considered are associated with a significant parameter, considering the same confidence level. The other indicators cannot be, therefore, considered as such, since the parameter that goes with them is not significant, and they do not help to measure the latent variable in question. The variables whose parameters were significant are those listed in Table 6. Taking this into account, a new model that considers only these indicators will be fitted.

After verifying that the identification of this second model, which will be called the reduced model to distinguish it from the general model, is possible, its forty-nine parameters have been estimated using the maximum–likelihood method.

**Table 3.** Estimates $\lambda^X$.

| Variable | Parameter | Estimate | Standard Error | *p*-Value |
|---|---|---|---|---|
| Museums | λ11 | −0.055 | 0.03 | 0.072 |
| Hotel Rooms | λ12 | 0.548 | 0.488 | 0.261 |
| Hotel-Apartment Rooms | λ13 | 0.646 | 0.601 | 0.282 |
| B&B and Guest House Rooms | λ14 | 0.01 | 0.046 | 0.831 |
| 4-key Apartment Rooms | λ15 | 0.015 | 0.011 | 0.154 |
| 3-key apartment Rooms | λ16 | 0.162 | 0.071 | 0.022 |
| 2-key apartment Rooms | λ17 | 0.045 | 0.061 | 0.462 |
| 1-key apartment Rooms | λ18 | 1.095 | 1.266 | 0.387 |
| Unemployment in Tourism | λ19 | 0.235 | 0.092 | 0.011 |
| Cinemas | λ110 | 0.057 | 0.047 | 0.228 |
| Film Screens | λ111 | 0.245 | 0.154 | 0.111 |
| Seating Capacity of Cinemas | λ112 | 0.167 | 0.088 | 0.058 |
| Banks | λ113 | 0.445 | 0.213 | 0.037 |
| Savings Banks | λ114 | 0.234 | 0.037 | 0 |
| Credit Unions | λ115 | 0.082 | 0.08 | 0.332 |
| Campsites | λ116 | −0.092 | 0.037 | 0.013 |
| Hotels | λ117 | 0.077 | 0.08 | 0.332 |
| Hotel-Apartment | λ118 | 0.163 | 0.103 | 0.112 |
| B&B and Guest Houses | λ119 | 0.066 | 0.046 | 0.153 |
| Restaurants | λ120 | −0.035 | 0.067 | 0.599 |
| Cafés | λ121 | −0.195 | 0.093 | 0.037 |
| Taxis | λ122 | 0.165 | 0.074 | 0.026 |
| Car and Driver Rental | λ123 | 0.026 | 0.045 | 0.563 |
| Ambulances | λ124 | −0.034 | 0.046 | 0.458 |
| Public Transp. + 10 travelers | λ125 | 0.14 | 0.084 | 0.098 |
| Public Transp. − 10 travelers | λ126 | 0.054 | 0.158 | 0.73 |
| Tax on Business Activities, Division 6 | λ127 | 0.346 | 0.057 | 0 |
| Tourism Index | λ128 | 0.254 | 0.118 | 0.032 |
| Tourist offices | λ129 | −0.069 | 0.041 | 0.09 |

Source: author's own elaboration.

**Table 4.** Estimates $\lambda^Y$.

| Variable | Parameter | Estimate | Standard Error | *p*-Value |
|---|---|---|---|---|
| Population census | λ11 | 0.001032976 | 0.000363793 | 0.002157952 |
| Municipal Register of Inhabitants | λ12 | 0.000586931 | 0.00015831 | 0.001134552 |
| Natural Population Growth | λ13 | 0.001377011 | 0.000666967 | 0.038962185 |
| Annual Personal Income Tax | λ14 | 0.00095996 | 0.000651886 | 0.140861746 |
| Registered Unemployment | λ15 | 0.000584655 | 0.000210245 | 0.005421972 |
| Electricity Consumption | λ16 | 0.000791828 | 0.000262599 | 0.00256685 |
| Private Cars | λ17 | 0.000655952 | 0.00023137 | 0.004581422 |
| Motorcycles | λ18 | 0.00120808 | 0.000493419 | 0.014349697 |
| Vans and Lorries | λ19 | 0.000401246 | 0.000114135 | 0.000438863 |
| Buses | λ110 | 0.000431626 | 0.000283032 | 0.127256758 |
| Commercial Vehicles | λ111 | 0.001238299 | 0.000974178 | 0.203685158 |
| Other Vehicles | λ112 | 0.000573089 | 0.000241057 | 0.017435037 |
| Buildings | λ113 | 0.000768413 | 0.000771245 | 0.3190913 |
| Real Estate | λ114 | 0.000726178 | 0.000218198 | 0.000874512 |
| Tax on Business Activities | λ115 | 0.000851517 | 0.00027342 | 0.001843695 |
| Patents | λ116 | $8.83129 \times 10^{-5}$ | 0.00012078 | 0.464665033 |
| Utility Models | λ117 | 0.000134803 | 0.000113754 | 0.23600248 |
| CNAE Establishments | λ118 | 0.000527962 | 0.000166179 | 0.001487706 |
| Economic Activity Index | λ119 | 0.000259519 | 0.000107521 | 0.015793131 |
| Population Older than 65 | λ120 | 0.00044647 | 0.000144067 | 0.001941418 |
| Foreign Population | λ121 | 0.010826384 | 0.014624298 | 0.459117307 |
| Marriages | λ122 | 0.000379619 | 0.000178742 | 0.033683876 |
| Health Centers | λ123 | $2.80269 \times 10^{-5}$ | 0.000117436 | 0.811371815 |

**Table 4.** *Cont.*

| Variable | Parameter | Estimate | Standard Error | *p*-Value |
|---|---|---|---|---|
| Vaccination Sites | λ124 | $2.06544 \times 10^{-5}$ | $8.55634 \times 10^{-5}$ | 0.809250385 |
| Local Clinics | λ125 | $3.24287 \times 10^{-5}$ | $6.96746 \times 10^{-5}$ | 0.641623011 |
| Public Education Institutions | λ126 | 0.000391141 | 0.000172887 | 0.023672504 |
| Public Libraries | λ127 | $3.32006 \times 10^{-5}$ | 0.000100095 | 0.740123875 |
| Pharmacies | λ128 | 0.00023677 | 0.000121739 | 0.051787404 |
| Peripherals Specialty Centers | λ129 | $3.43729 \times 10^{-5}$ | $2.71189 \times 10^{-5}$ | 0.20498026 |
| Public Hospitals | λ130 | $6.94556 \times 10^{-5}$ | $7.37034 \times 10^{-5}$ | 0.346004978 |
| Private Hospitals | λ131 | $4.6907 \times 10^{-5}$ | $3.92853 \times 10^{-5}$ | 0.232475151 |
| Public Hospital Beds | λ132 | 0.000114082 | $8.5928 \times 10^{-5}$ | 0.184296835 |
| Private Hospital Beds | λ133 | $-1.26812 \times 10^{-5}$ | 0.000197234 | 0.948735066 |

Source: author's own elaboration.

**Table 5.** Estimate of parameter $\gamma$.

| Estimate | Standard Error | *p*-Value |
|---|---|---|
| 658.694 | 77.914 | 0 |

Source: author's own elaboration.

**Table 6.** Indicators with parameters significantly different from zero.

| Tourism Development | Socioeconomic Development |
|---|---|
| 3 Key Apartment rooms | Population Census |
| Unemployment in Tourism | Municipal Register of Inhabitants |
| Banks | Natural Population Growth |
| Savings Banks | Population older than 65 |
| Campsites | Tax on Business Activities |
| Cafés | Electricity Consumption |
| Tax on Business Activities, Division 6 | CNAE Establishments |
| Tourism Index | Marriages |
| | Registered Unemployment |
| | Public Education Institutions |
| | Real Estate |
| | Private Cars |
| | Motorcycles |
| | Vans and Lorries |
| | Other Vehicles |
| | Economic Activity Index |

Source: author's own elaboration.

As can be seen, now all parameters $\lambda^{X'}$ (Table 7), $\lambda^{Y'}$ (Table 8) e$\gamma'$ (Table 9) are significantly different from zero at a 95% confidence level. This means that, on the one hand, given this confidence level, indicators $\lambda^{X'}$ and $\lambda^{Y'}$' reduce the initial set of indicators to those presented in Table 6, while, on the other hand, the relationship $\gamma'$ provides support for the causal relationship between tourism development and socioeconomic development.

**Table 7.** Estimates $\lambda^{X'}$.

| Variable | Parameter | Estimate | Standard Error | *p*-Value |
|---|---|---|---|---|
| 3 Key Apartment rooms | λ'1 | 0.158 | 0.067 | 0.019 |
| Unemployment in Tourism | λ'2 | 0.232 | 0.09 | 0.01 |
| Banks | λ'3 | 0.447 | 0.207 | 0.031 |
| Savings Banks | λ'4 | 0.23 | 0.037 | 0 |
| Campsites | λ'5 | −0.091 | 0.038 | 0.015 |
| Cafés | λ'6 | −0.198 | 0.094 | 0.035 |
| Tax on Business Activities, Div. 6 | λ'7 | 0.346 | 0.057 | 0 |
| Tourism Index | λ'8 | 0.249 | 0.113 | 0.028 |

Source: author's own elaboration.

**Table 8.** Estimates $\lambda^{Y'}$.

| Variable | Parameter | Estimate | Standard Error | *p*-Value |
|---|---|---|---|---|
| Population Census | $\lambda'1$ | 0.000517837 | 0.000128612 | $5.66498 \times 10^{-5}$ |
| Municipal Register of Inhabitants | $\lambda'2$ | 0.00029584 | $3.9067 \times 10^{-5}$ | 0 |
| Natural Population Growth | $\lambda'3$ | 0000193697 | $7.07068 \times 10^{-5}$ | 0.006154518 |
| Population older than 65 | $\lambda'4$ | 0.000703917 | 0.000267858 | 0.0085902 |
| Tax on Business Activities | $\lambda'5$ | 0.000233156 | $2.93006 \times 10^{-5}$ | 0 |
| Electricity Consumption | $\lambda'6$ | 0.000432308 | $9.39531 \times 10^{-5}$ | $4.19841 \times 10^{-6}$ |
| CNAE Establishments | $\lambda'7$ | 0.000400921 | $9.15103 \times 10^{-5}$ | $1.1805 \times 10^{-5}$ |
| Marriages | $\lambda'8$ | 0.00026756 | $5.2749 \times 10^{-5}$ | 0 |
| Registered Unemployment | $\lambda'9$ | 0.000294285 | $6.78383 \times 10^{-5}$ | $1.43767 \times 10^{-5}$ |
| Public Education Institutions | $\lambda'10$ | 0.000198861 | $6.77493 \times 10^{-5}$ | 0.00333278 |
| Real Estate | $\lambda'11$ | 0.000366228 | $7.01072 \times 10^{-5}$ | 0 |
| Private Cars | $\lambda'12$ | 0.000333008 | $8.44424 \times 10^{-5}$ | $8.02632 \times 10^{-5}$ |
| Motorcycles | $\lambda'13$ | 0.000611977 | 0.000193562 | 0.001568699 |
| Vans and Lorries | $\lambda'14$ | 0.000202711 | $2.73106 \times 10^{-5}$ | 0 |
| Other Vehicles | $\lambda'15$ | 0.000288726 | $7.8953 \times 10^{-5}$ | 0.000255243 |
| Economic Activity Index | $\lambda'16$ | 0.000132523 | $4.17495 \times 10^{-5}$ | 0.001502328 |

Source: author's own elaboration.

**Table 9.** Estimate of parameter $\gamma'$.

| Estimate | Standard Error | *p*-Value |
|---|---|---|
| 1301.292 | 51.895 | 25.075 |

Source: author's own elaboration.

Once the parameters have been obtained, the goodness-of-fit of the reduced model is analyzed, comparing the results with those of the general model. To do this, we will use the measurements outlined in Table 10 as a basis.

**Table 10.** Goodness-of-fit measurements for the general and reduced models.

| | General Model | Reduced Model |
|---|---|---|
| **Chi-Square (*p*-Value)** | **4075.861(0.000)** | **712.307(0.000)** |
| *NFI* | *0.294* | *0.69* |
| *NNFI* | *0.401* | *0.749* |
| *CFI* | *0.421* | *0.772* |
| *IFI* | *0.431* | *0.775* |
| *MFI* | *0.000* | *0.193* |
| *GFI* | *0.537* | *0.682* |
| *AGFI* | *0.505* | *0.62* |
| AIC | 20,630.206 | 5748.216 |
| BIC | 20,997.912 | 5892.357 |
| RMR | 1.754 | 0.078 |

Source: author's own elaboration.

In general terms, we can conclude that the reduced model improves the measures of fit of the general model. In both models, the hypothesis that the observed covariance matrix is equal to the reproduced covariance matrix is rejected. Although this may be due to the fact that the model does not adequately reproduce the covariance matrix, this test is severely affected by large sample sizes, as in this case. Moreover, the reduced model improves all relative goodness-of-fit measurements (shown in italics in Table 10), obtaining values closer to the unit. The reduced model is associated also with smaller values of AIC, BIC and RMR compared to the same values for the general model, which implies a better fit of the former model compared to the latter [43–48].

### 4.1. Ranking of Municipalities Based on Tourism Development and Socioeconomic Development

Finally, municipalities will be classified into two categories: the first one will be based on the value of the tourism development index that each place presents, while the second one will be based on the value of the socioeconomic development index. As is well known,

these indexes are not directly observable or measurable, so in order to obtain their value in each of the one hundred and forty municipalities that make up the set of observations, the technique known as principal component analysis has been used.

4.1.1. Tourism Development Index

By applying this statistic technique to this specific case, and in order to obtain the values of the tourism development index (TDI), four principal components have been extracted from the correlation matrix (as the variables showed very different variances), which can be considered subindexes or sub-measures of that index. It will be calculated then as follows:

$$TDI = w_1 * CP_1^{Tur} + w_2 * CP_2^{Tur} + w_3 * CP_3^{Tur} + w_4 * CP_4^{Tur}$$

where $w_1$, $w_2$, $w_3$, $w_4$ weight each subindex according to the percentage of variance that explains each one of them. Thus, the first four principal components explain, together, 70.049% of the total variability of the observations, which is distributed as presented in Table 11.

**Table 11.** Percentage of variance explained by the four main principal components of the tourism development index (TDI).

| Component (w) | % of Explained Variance | Accumulated % of Explained Variance | % over the Total Explained Variance of the Four Components |
|---|---|---|---|
| 1 | 29.118 | 29.118 | 41.57 |
| 2 | 14.804 | 43.922 | 21.13 |
| 3 | 14.066 | 57.988 | 20.08 |
| 4 | 12.061 | 70.049 | 17.22 |

Source: author's own elaboration.

Table 12 allows for obtaining the expression of the four extracted components according to the dummy variables, as was already done with the principal components for the TDI.

**Table 12.** Weight of the variables in the components of TDI.

| | Comp. 1 | Comp. 2 | Comp. 3 | Comp. 4 |
|---|---|---|---|---|
| 3 Key Apartment Rooms | 0.355608 | 0.308498 | −0.237041 | −0.52398 |
| Unemployment in Tourism | 0.420598 | 0.15782 | 0.342407 | 0.0717237 |
| Banks | 0.301068 | −0.52197 | −0.0648848 | 0.387813 |
| Savings Banks | 0.533027 | 0.244683 | −0.0922956 | −0.180926 |
| Campsites | −0.216875 | 0.0449372 | −0.633815 | −0.0777056 |
| Cafés | −0.0626185 | 0.621905 | 0.290455 | 0.456186 |
| Tax on Business Activities, Div. 6 | 0.507244 | −0.266628 | −0.0871499 | 0.165784 |
| Tourism Index | 0.116742 | 0.296181 | −0.565827 | 0.543635 |

Source: author's own elaboration.

Once the weight of the TDI variables is known, the value of the TDI for each city is calculated. The thirty municipalities with a higher level of tourism development are listed in Table 13.

**Table 13.** Municipalities with a higher level of tourism development.

| Municipality | Comp. 1 | Comp. 2 | Comp. 3 | Comp. 4 | IDT |
|---|---|---|---|---|---|
| Bormujos | 5.88527 | 3.93266 | 1.93981 | 2.0685 | 402.3187345 |
| Mairena del Aljarafe | 1.91664 | 4.35771 | 0.69353 | 1.91019 | 218.5726913 |
| Alhaurín de la Torre | 3.4297 | −0.972844 | 2.37017 | 0.627402 | 180.4133113 |
| Cártama | 3.03012 | 1.96044 | −0.750701 | 1.33657 | 175.3278449 |
| Espartinas | 2.48158 | 1.59249 | 0.92183 | 0.843651 | 169.8466109 |
| Mijas | 5.8366 | −2.09041 | 0.609789 | −2.61554 | 165.662063 |
| Arcos de la Frontera | −0.126614 | −3.02664 | 6.76649 | 5.03106 | 153.2897252 |
| Manilva | 4.31222 | −0.141974 | −2.36116 | 1.21785 | 149.818359 |
| Rincón de la Victoria | 4.76305 | −1.2035 | 0.678887 | −2.32184 | 146.2199997 |
| Níjar | 1.33381 | −0.0133497 | 1.29364 | 1.7403 | 111.1086597 |
| Conil de la Frontera | 3.36054 | −1.73103 | 0.990885 | −1.33429 | 100.0414809 |
| Vera | 3.112 | −0.14325 | −1.57421 | 0.234547 | 98.76773004 |
| Churriana de la Vega | 2.17384 | 0.279242 | 0.22043 | −0.186603 | 97.479843 |
| Gabias, Las | 1.59008 | 0.947594 | 0.10813 | 0.238493 | 92.40038668 |
| Ogíjares | 1.43424 | 1.01191 | 0.00758821 | 0.0713328 | 82.38373717 |
| Estepona | 2.7401 | −0.86931 | −0.0646837 | −0.893073 | 78.85987094 |
| Atarfe | 0.67175 | 1.03792 | 0.258291 | 1.229 | 76.20576038 |
| Huércal de Almería | 1.994 | 0.239576 | −0.703257 | 0.13649 | 76.18177812 |
| Cartaya | 1.07111 | −0.40103 | 0.492728 | 1.62201 | 73.87726924 |
| Carlota, La | 0.539684 | 1.1927 | 0.00149045 | 0.520215 | 56.62444542 |
| Torrox | 1.69041 | −0.920938 | 0.63855 | −0.540539 | 54.32492618 |
| Chiclana de la Front. | 1.50369 | −1.19358 | −0.830065 | 1.92892 | 53.8363451 |
| Vícar | 0.726883 | 0.8573 | −0.0605706 | 0.227707 | 51.0361322 |
| Zubia, La | 1.04293 | 0.438445 | 0.229302 | −0.51883 | 48.28907451 |
| Roquetas de Mar | 2.02715 | −1.24618 | −0.0363006 | −0.563562 | 47.50338841 |
| Vejer de la Frontera | 0.453213 | 1.05185 | −0.934827 | 1.12301 | 41.63256095 |
| Gines | 0.576762 | 1.01377 | −0.25752 | −0.0489823 | 39.38247963 |
| Punta Umbría | 1.23533 | −0.222054 | −0.284434 | −0.105057 | 39.14015082 |
| Isla Cristina | −0.288333 | 0.212908 | 1.03192 | 1.42605 | 37.79027783 |
| Tomares | 0.34695 | 0.794354 | 0.347139 | −0.0252626 | 37.74294067 |

Source: author's own elaboration.

On the basis of the data obtained, it is possible to identify three main groups of medium-sized cities according to their spatial characteristics, resources and location. Firstly, we find a group referred to as "synergistic medium-sized cities", including those cities that are part of the metropolitan area of Andalusian provincial capitals, which in principle, according to the position of many of them in the ranking, seem to have a strong tourist appeal. However, these kinds of cities are not in line with the concept of tourism destination, as their relevance is due to the fact that they offer a wide variety of services at competitive prices, which makes them become dormitory medium-sized cities linked to any of the big tourism capitals of Andalusia (Seville, Malaga and Granada).

In this analysis, this group of cities represents 25% of all medium-sized cities of Andalusia. Among the thirty cities that have shown a higher level of tourism development, we find the municipalities of Bormujos, Mairena del Aljarafe, Espartinas, Churriana de la Vega, Las Gabias, Ogíjares, Atarfe, Huércal de Almeria, Cartaya, La Carlota, Chiclana de la Frontera, La Zubia, Gines, Punta Umbría and Tomares. These cities represent 50% of the thirty municipalities with a higher level of tourism development.

A second group, referred to as "coastal medium-sized cities", has also been identified. It includes those cities that meet the definition of coastal tourism destinations, as they have high tourist appeal and possess a variety of natural tourism resources typical of the Andalusian coast. In the sample, this group represents 19% of the one hundred and forty observations. Out of the thirty municipalities with a higher level of tourism development, this group comprises the municipalities of Mijas, Marbella, Rincón de la Victoria, Níjar, Conil de la Frontera, Vera, Estepona, Torrox, Roquetas de Mar and Isla Cristina, representing 34% of the sample.

The third group that has been identified corresponds to "inland medium-sized cities", which includes those cities located in the interior classified as tourism destinations. These cities have a historical and cultural heritage located in their territorial space that makes them a unique tourism site. This group represents 56% of all municipalities analyzed. As seen in Table 13, it includes cities such as Alhaurín de la Torre, Cártama, Arcos de la Frontera, Vícar and Vejer de la Frontera, which account for 16% of the more developed municipalities from the tourism point of view.

4.1.2. Socioeconomic Development Index

In line with the process carried out with the dummy variables of tourism development, five principal components have been extracted from the correlation matrix (since the variables show very different variances), which can be considered subindexes or sub-measures of the socioeconomic development index (SDI). It is calculated as follows:

$$S = w_1 * CP_1^{Soc} + w_2 * CP_2^{Soc} + w_3 * CP_3^{Soc} + w_4 * CP_4^{Soc} + w_5 * CP_5^{Soc}$$

where $w_1, w_2, w_3, w_4, w_5$ weight each subindex according to the percentage of variance that explains each one of them. Thus, the first four principal components explain, together, 73.303% of the total variability of the observations.

The values included in the fourth column of Table 14 will play the role of weights to calculate the SDI.

**Table 14.** Percentage of variance explained by the five main principal components of socioeconomic development index (SDI).

| Component (w) | % of Explained Variance | Accumulated % of Explained Variance | % over the Total Explained Variance of the Five Components |
|---|---|---|---|
| 1 | 46.478 | 46.478 | 63.40 |
| 2 | 7.425 | 53.904 | 10.13 |
| 3 | 6.817 | 60.721 | 9.30 |
| 4 | 6.670 | 67.391 | 9.10 |
| 5 | 5.912 | 73.303 | 8.07 |

Source: author's own elaboration.

Meanwhile, Table 15 shows the weight of each variable on each one of the five components extracted. Its content enables the expression of each component, so, for instance, the first one of them can be calculated considering the following expression:

$$
\begin{aligned}
CP_1^{Soc} = {} & 0.059891*CensusPopulation + 0.342307 * MunicipalRegistrer + 0.134798 \\
& *NaturalPopGrowth + 0.262143 * PopOlder65 + 0.341386 * TaxBusiness + 0.237496 \\
& *Electricity + 0.289705 * Establishment + 0.254282 * Marriages + 0.194438 \\
& *RegistreredUnemployment + 0.24851 * PublicEducationInstitutions + 0.252791 \\
& *RealEstate + 0.313012 * Motorclycles + 0.298963 * Vans\ and\ Lorries + 0.125749 \\
& *OtherVechicles + 0.142935 * EconomicActivityIndex
\end{aligned}
$$

where the values of the variables have been previously standardized. Similarly, the other four components can be calculated.

**Table 15.** Weight of the variables in the components of SDI.

|  | Comp. 1 | Comp. 2 | Comp. 3 | Comp. 4 | Comp. 5 |
|---|---|---|---|---|---|
| Population Census | 0.059891 | −0.327018 | −0.50011 | −0.428438 | −0.4939 |
| Municipal Register of Inhabitants | 0.342307 | −0.148098 | 0.0213024 | 0.100488 | −0.0545186 |
| Natural Population Growth | 0.134798 | −0.103781 | 0.443147 | −0.393751 | 0.35976 |
| Population older than 65 | 0.262143 | −0.190393 | −0.200444 | 0.225541 | −0.0589534 |
| Tax on Business Activities | 0.341386 | 0.137542 | −0.000866876 | 0.0431962 | −0.0417789 |
| Electricity Consumption | 0.237496 | 0.207661 | 0.0526683 | 0.226513 | 0.106745 |
| CNAE Establishments | 0.289705 | 0.0475271 | 0.0340754 | 0.0251954 | −0.146375 |
| Marriages | 0.254282 | −0.197363 | 0.264847 | −0.192846 | 0.013248 |
| Registered Unemployment | 0.194438 | −0.409 | 0.0450327 | 0.528832 | 0.103829 |
| Public Education Institutions | 0.24851 | 0.387668 | 0.118448 | −0.214531 | −0.201692 |
| Real Estate | 0.252791 | −0.152705 | 0.0879963 | −0.0297825 | −0.0612276 |
| Private Cars | 0.313012 | 0.051955 | 0.0910758 | −0.0834563 | −0.0392913 |
| Motorcycles | 0.298963 | 0.0410904 | 0.0534335 | −0.260765 | −0.0520704 |
| Vans and Lorries | 0.289393 | −0.119171 | −0.309637 | 0.054648 | 0.205396 |
| Other Vehicles | 0.125749 | 0.1498 | −0.532566 | −0.226192 | 0.66911 |
| Economic Activity Index | 0.142935 | 0.582184 | −0.156035 | 0.23184 | −0.189965 |

Source: author's own elaboration.

Using the expressions of the principal components, it is possible to calculate the scores of each of the observations of the sample data for each factor to, in turn, obtain the value of the SDI in each municipality. It can be verified, therefore, that the five municipalities with a higher SDI are those listed in Table 16.

In order to draw conclusions regarding the SDI, it is necessary to compare the results with those discussed above, in the ranking of tourist cities (Table 13). The joint analysis of Tables 13 and 16 allows for observing that some municipalities appear in the tables as changing their position, or even do not appear in any of these two tables. This situation is due, according to the variables used in the analysis, to the fact that there are some municipalities with a high level of socioeconomic development which, however, does not correspond to the same level of tourism development; or, on the contrary, there are municipalities with a high level of tourism development which, from a socioeconomic perspective, shows a lower level of development.

In short, it can be concluded that the reason for this disparity lies in the use of tourism as a development factor or, on the contrary, in the use of other factors that obviate the possible potential of tourism, as there may be other factors, not related to tourism, that determine, to a greater extent, the economic development level.

**Table 16.** Municipalities with higher levels of socioeconomic development.

| Municipality | Comp. 1 | Comp. 2 | Comp. 3 | Comp. 4 | Comp. 5 | SDI |
|---|---|---|---|---|---|---|
| Bormujos | 14.69 | 5.23844 | 0.506293 | −1.98546 | −1.51406 | 958.8337719 |
| Espartinas | 13.869 | 0.526699 | 3.20836 | −3.74007 | −0.207443 | 878.7591069 |
| Manilva | 7.92308 | −0.712755 | −0.652149 | 2.11565 | 0.0884851 | 509.0045679 |
| Gabias, Las | 6.72357 | −1.65517 | −0.277091 | −0.188444 | 1.80759 | 419.8029305 |
| Aljaraque | 5.55479 | −0.727955 | 0.759785 | −0.956794 | −0.172061 | 341.7701447 |
| Huércal de Almería | 5.49384 | −2.14841 | 1.61711 | 0.729305 | −1.24241 | 338.1956125 |
| Vera | 4.9062 | −0.192117 | 0.23202 | 2.56109 | −0.590159 | 329.8080567 |
| Rincón de la Victoria | 5.06948 | 0.565324 | −1.8968 | 0.412354 | 0.821381 | 319.8724902 |
| Mijas | 5.01517 | −0.122563 | −1.96423 | 0.745635 | 1.17795 | 314.7442108 |
| Alhaurín de la Torre | 4.68136 | −0.655622 | −1.15707 | 0.598437 | −0.04062 | 284.5139954 |
| Churriana de la Vega | 4.51909 | −1.44647 | 0.0774826 | −0.384055 | 1.06049 | 277.6414069 |
| Benalmádena | 4.05804 | −1.37666 | −0.986508 | 0.504352 | 0.289838 | 241.0882417 |
| Roquetas de Mar | 3.73454 | −0.503804 | −0.177897 | 1.52535 | −0.666496 | 238.5139217 |
| Cártama | 3.52839 | −1.0864 | −0.971185 | 0.487846 | 2.0657 | 224.7722711 |
| Ogíjares | 3.33218 | 0.56183 | −1.39802 | 1.36463 | −0.107173 | 215.5032108 |
| Níjar | 2.31592 | −0.322198 | 0.0884227 | 0.602573 | 0.568754 | 154.4610525 |
| Estepona | 2.20681 | 0.10605 | −1.03165 | 1.01683 | 0.493294 | 144.6257311 |
| Carlota, La | 1.54081 | 0.474023 | 1.16246 | −1.74534 | 2.84786 | 120.3997212 |
| Vícar | 1.70738 | −0.577681 | 0.711705 | 1.60112 | −0.963296 | 115.8112333 |
| Chiclana de la Frontera | 1.38399 | 2.10855 | −0.913349 | 1.86472 | −0.730836 | 111.6815373 |
| Cartaya | 1.43233 | 0.986603 | 0.831541 | 0.783642 | −0.765644 | 109.4897368 |
| Atarfe | 1.67736 | −1.41822 | 2.09083 | 0.0873945 | −0.515121 | 108.0610379 |
| Tomares | 1.33003 | 2.90406 | −0.263235 | 0.476293 | −1.56187 | 103.0239197 |
| Torrox | 1.65945 | −0.0253001 | −0.87963 | 0.33133 | 0.280097 | 102.0477668 |
| Armilla | 1.48759 | −1.1329 | 0.978402 | −0.428209 | −0.0125158 | 87.93836319 |
| Guillena | 1.31926 | −1.42037 | 1.2345 | 0.0419533 | 0.668946 | 86.51375515 |
| Cuevas del Almanzora | 0.932843 | 0.151803 | 0.258036 | 2.04567 | 0.489846 | 85.64839961 |
| Conil de la Frontera | 1.47518 | −1.10451 | −0.233657 | −0.863921 | 0.573971 | 76.93498047 |
| Mairena del Aljarafe | 1.22756 | 0.245088 | −0.482732 | 0.253832 | −0.528858 | 73.86262498 |
| Torremolinos | 1.29246 | −0.104411 | −0.135436 | −0.561609 | −0.192187 | 72.96313478 |

Source: author's own elaboration.

### 4.1.3. Global Index

A general index of tourism and economic development has been also elaborated, by averaging the values obtained for the two indexes. The thirty municipalities leading the general index are shown in Table 17.

Table 17 presents the ranking of cities by means of a global index representing the sum of the components that mark TDI and SDI. Therefore, the overall result shows major tourism destinations located both in coastal and inland Andalusia (shown in italics in the table), while the others correspond to medium-sized cities within the metropolitan area of large provincial capitals.

As a final discussion, it is worth mentioning that the direct relationship between tourism development and socioeconomic development is a matter of importance, especially due to the current situation generated by the COVID-19 pandemic in which the main world economies, such as the Spanish economy, linked to tourism have suffered a greater drop in its indicators of socioeconomic development.

Furthermore, the established ranking makes it possible to clearly see that those cities near the coast or large provincial capitals develop with a very clear pattern that corresponds to medium-sized cities in an area of touristic importance in Andalusia. Some things to bear in mind are some measures included in the conclusions.

**Table 17.** Municipalities with the highest tourism–socioeconomic development.

| Municipalities | Socioeconomic Development Index (SDI) | Tourism Development Index (TDI) | General Index |
|---|---|---|---|
| Bormujos | 958.8337719 | 402.3187345 | 680.5762532 |
| Espartinas | 878.7591069 | 169.8466109 | 524.3028589 |
| *Manilva* | 509.0045679 | 149.818359 | 329.4114634 |
| Gabias, Las | 419.8029305 | 92.40038668 | 256.1016586 |
| *Mijas* | 314.7442108 | 165.662063 | 240.2031369 |
| *Rincón de la Victoria* | 319.8724902 | 146.2199997 | 233.0462449 |
| *Alhaurín de la Torre* | 284.5139954 | 180.4133113 | 232.4636534 |
| *Vera* | 329.8080567 | 98.76773004 | 214.2878934 |
| Huércal de Almería | 338.1956125 | 76.18177812 | 207.1886953 |
| Cártama | 224.7722711 | 175.3278449 | 200.050058 |
| Churriana de la Vega | 277.6414069 | 97.479843 | 187.5606249 |
| Aljaraque | 341.7701447 | 0.87402465 | 171.3220847 |
| Ogíjares | 215.5032108 | 82.38373717 | 148.943474 |
| Mairena del Aljarafe | 73.86262498 | 218.5726913 | 146.2176581 |
| *Roquetas de Mar* | 238.5139217 | 47.50338841 | 143.008655 |
| *Níjar* | 154.4610525 | 111.1086597 | 132.7848561 |
| *Benalmádena* | 241.0882417 | 17.42967523 | 129.2589584 |
| *Estepona* | 144.6257311 | 78.85987094 | 111.742801 |
| Atarfe | 108.0610379 | 76.20576038 | 92.13339913 |
| Cartaya | 109.4897368 | 73.87726924 | 91.68350303 |
| Carlota, La | 120.3997212 | 56.62444542 | 88.5120833 |
| *Conil de la Frontera* | 76.93498047 | 100.0414809 | 88.48823069 |
| Vícar | 115.8112333 | 51.0361322 | 83.42368273 |
| *Chiclana de la Frontera* | 111.6815373 | 53.8363451 | 82.75894119 |
| *Torrox* | 102.0477668 | 54.32492618 | 78.18634648 |
| *Arcos de la Frontera* | −11.9905583 | 153.2897252 | 70.64958346 |
| Tomares | 103.0239197 | 37.74294067 | 70.38343018 |
| *Cuevas del Almanzora* | 85.64839961 | 14.25678111 | 49.95259036 |
| Guillena | 86.51375515 | 9.629193718 | 48.07147443 |
| Peligros | 58.86347768 | 36.13862296 | 47.50105032 |

Source: author's own elaboration.

## 5. Conclusions

The technique of structural equation modeling was applied to a total of one hundred and forty observations, for which a total of sixty-two relative growth rates were measured, obtained from the measurement of other many features in two different time periods. Twenty-nine of the relative rates of change analyzed make up the group of indicators of a latent variable that has been called "tourism development", while the remaining thirty-three form a group of indicators related to another latent variable, referred to as "socioeconomic development".

The maximum likelihood estimation of the parameters of the structural equation model revealed the existence of many non-significant parameters, so a re-specification of the model was performed by eliminating those variables whose parameters could be considered zero.

As a result, we obtained the significance of most of the parameters of the model at a 95% confidence level, and the support to the hypothesis of causality between tourism development and socioeconomic development, at that same confidence, which is especially relevant, taking into account the current situation of the COVID-19 pandemic and the relationship between those territories that have experienced a decrease in tourism.

Once the ranking of municipalities was obtained, using the analysis of principal components, three lists were elaborated (municipalities with more tourism development, municipalities with more socioeconomic development and municipalities with more tourism and socioeconomic development) which allow for drawing the necessary conclusions for the set hypothesis.

In fact, this research work has demonstrated that, in the cities analyzed, there is a relationship between tourism development and socioeconomic development, or rather, that tourism development influences socioeconomic development. Furthermore, these cities develop with a very clear pattern; they grow around key tourist development areas for the Andalusia region.

It has also been shown that this relationship does not occur with equal intensity in all cities. In fact, it has been found that cities leading the ranking of the TDI do not occupy the same position in the ranking of the SDI, and the other way round, which means that, even having demonstrated this causal relationship, it is conditioned by a number of factors that make it more or less intense.

The next step, and therefore, a future line of research, would be to identify those factors that help or hinder this relationship, which ultimately explain why the position occupied by the cities in the two rankings is not the same.

**Funding:** This research received no external funding.

**Institutional Review Board Statement:** Not applicable.

**Informed Consent Statement:** Informed consent was obtained from all subjects involved in the study.

**Conflicts of Interest:** The authors declare no conflict of interest.

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
