# Peer review of "Does the Tourism Development of a Destination Determine Its Socioeconomic Development? An Analysis through Structural Equation Modeling in Medium-Sized Cities of Andalusia, Spain"

_land, doi:10.3390/land10040378_

Round 1

Reviewer 1 Report

The article is well-organised and the research methods have been properly used and presented. Good use of the SEM analysis to search for relations between the development of tourism and economic development of analysed cities. A particular advantage of the article is the distinction of cities into three categories as well as the indication of potential directions for future research. It would be worth expanding the literature with more up-to-date entries.

Author Response

Dear Reviewer,

Thank you very much for your considerations.

Sincerely,

Reviewer 2 Report

The work is a good one, and the approach is interesting where the analysis is well conduct. However, some changes need to be considered to improve the methodology. First of all, the authors should better define their initial data: are there some missing data? What is the quality of the data? The sample collected should be correct for the inference to be made. The real sample collected from the authors must be related to the problem considered, and the quality of the data is maximized. At the same time, the term "causality" is too strong. There are surely important elements in this sense, but there are could be problems with confounding factors and so on. So the word "causality," used in work, should be carefully used in this case. Finally, there is the need for sensitivity analyses to robustify the results. At the same time, the authors should evaluate and emphasize the different validation indexes. Finally, the work needs a careful revision of the English language. Finally, the authors should specify the computational approaches used (what software they used on the analysis and the procedures followed).

Author Response

Dear Reviewer,

Changes have been incorporated into the article following its instructions. In the manuscript you can see the improvements and modifications in red. English language has been improved and the software has been added. In addition, both the introduction and the conclusions have been restructured and methodology has been more clearly specified. More current references have also been incorporated in order to contribute to the subject of the article. I hope that in this review process the article presented is in accordance with your indications in the review.

Thank you very much for your attention.  I look forward to hearing from you.

Sincerely,

Reviewer 3 Report

Dear Authors,

Thank you for the opportunity to read and review your manuscript submitted to Land. After reading the manuscript I am convinced that the research of the relationship between tourism development and socioeconomic development in medium-sized cities is relevant, instrumental and valuable. It is praiseworthy that you have chosen a rather long period in your research. A list of indicators is also impressive. However, I offer the following recommendations that may improve the quality of the paper:

  1. The introduction needs to be restructured. In the current version of the introduction you concentrate on cities and their importance for socioeconomic development. As far as I understand, the object of the research is the relationship between tourism development and socioeconomic development in medium-sized cities, therefore the introduction should clearly present why it is relevant to research this relationship in case of medium-sized cities.
  2. The same terms should be used all over the manuscript. The topic indicates that the context of the research are medium-sized cities, while in line 74 and 440 you use the other term (town).
  3. The references should be specified according to the requirements of the journal. For example, in line 83 you mention “For [22], the process of ….”. It is unclear – for who? In line 85 you mention “this author…”. It is unclear – which author?
  4. The literature research should be expanded. It covers only tourism as an endogenous development tool, while there is no scientific background for the socioeconomic development and relationship between tourism development and socioeconomic development.
  5. The title of 2.2 section is unclear. Why you use “versus”, for the research that intends to find out the relationship between the two variables?
  6. The content of 2.2 section is fragmentary and inconsistent. It sounds like a presentation of the object of empirical research, while the title specifies other things.
  7. “Some authors” mentioned in line 117 should be specified.
  8. In lines 141-144 you have mentioned “characteristics that may be considered”, however in your research you have considered (specified) only one of them: population size. What about the others? In lines 153-155 you have mentioned other important features? Why are they mentioned if they are not considered for the cities under the study? Such a list of indicators without the data creates a confusion: it is unclear which exact meanings of the indicators let to identify city as the medium-sized?
  9. Even though the list of variables used to measure tourism development and socioeconomic development is impressive, it lacks scientific background. It remains unclear why these exact variables were chosen? How do I know that they represent for example tourism development component? Therefore, I recommend to strengthen the scientific background of the chosen indicators. Units of measurement of every indicator also should be mentioned.
  10. What is the logic of global index? Does it have any scientific background?
  11. Discussion section is missing.
  12. Conclusions present mainly technical results. It is recommended to specify how does the research contribute to the development of tourism of medium-sized cities. What do the results mean for the socioeconomic situation in these cities?
  13. Under the conditions of COVID-19 the tourism in Spain is the segment that has suffered painfully since March 2020. I clearly understand that the period of your research does not cover 2020, however in introduction and in discussion part it is valuable to provide your insights about the current context. In fact, the current situation adds extra relevance to your research.

Once again, thank you for this opportunity and I hope that these suggestions will be helpful in improving the manuscript. I look forward to reading the revised version.

Author Response

Dear Reviewer,

Thank you very much for your instructions. These changes have been incorporated into the article following your valuable recommendations. In the manuscript you can see the improvements and changes in red.

In order to answer your instructions, here you can check point by point:

  1. The introduction needs to be restructured. In the current version of the introduction you concentrate on cities and their importance for socioeconomic development. As far as I understand, the object of the research is the relationship between tourism development and socioeconomic development in medium-sized cities, therefore the introduction should clearly present why it is relevant to research this relationship in case of medium-sized cities.

Improvements have been incorporated following your valuable instructions, taking into account that medium-sized cities represent a considerable field to study tourism development and its relationship with economic development, as you suggest, the research object with the purpose the introduction and addition of tourism-related bibliography as a tool for socio-economic development in medium-sized cities has been restructured.

Also, these are the papers that have been updated:

[12] Hidalgo, B. D. E. (2020). Patrimonio cultural como factor de desarrollo territorial resiliente en áreas rurales. El caso de Mértola (Portugal). PASOS Revista de Turismo y Patrimonio Cultural2020, 18(1), 9-25.

[13] González Reverté, F., & Blay Boqué, J. La atracción migratoria de las ciudades turísticas y la transformación del sistema urbano litoral mediterráneo español. Un análisis a escala local del período 1991 a 2011. Revista de Estudios Regionales, 2019, 114, pp. 171-196.

[14] Mass, W. Costa Ibérica, MVRDV Costa Ibérica. Hacia la ciudad del ocio, Barcelona, 2001, ACTAR , pp. 70-99.

[15] Stock, M. & Lucas, L. La double révolution urbaine du tourisme, Espaces et sociétés, 2012, 151, pp. 15-30.

[16] González, F. & Anton, S. The formation of tourism cities and their effect on the Spanish Mediterranean urban system”, comunicación presented to Regional Studies Association Congress Evolution and transformation in tourism destinations: Revitalisation through innovation?, 2014, Vila-seca.

2. The same terms should be used all over the manuscript. The topic indicates that the context of the research are medium-sized cities, while in line 74 and 440 you use the other term (town).

Improvements have been incorporated following your valuable instructions

3. The references should be specified according to the requirements of the journal. For example, in line 83 you mention “For [22], the process of ….”. It is unclear – for who? In line 85 you mention “this author…”. It is unclear – which author?

Improvements have been incorporated following your valuable instructions

4. The literature research should be expanded. It covers only tourism as an endogenous development tool, while there is no scientific background for the socioeconomic development and relationship between tourism development and socioeconomic development.

Improvements have been incorporated following your valuable instructions.  For this, historical antecedents of the evolution of tourism have been added as a socio-economic development strategy. I have updated this section with the paper: González Reverté, F., & Blay Boqué, J. La atracción migratoria de las ciudades turísticas y la transformación del sistema urbano litoral mediterráneo español. Un análisis a escala local del período 1991 a 2011. Revista de Estudios Regionales, 2019, 114, pp. 171-196.

5. The title of 2.2 section is unclear. Why you use “versus”, for the research that intends to find out the relationship between the two variables?

Improvements have been incorporated following your valuable instructions

6. The content of 2.2 section is fragmentary and inconsistent. It sounds like a presentation of the object of empirical research, while the title specifies other things.

Improvements have been incorporated following your valuable instructions

7. “Some authors” mentioned in line 117 should be specified.

Improvements have been incorporated following your valuable instructions

8. In lines 141-144 you have mentioned “characteristics that may be considered”, however in your research you have considered (specified) only one of them: population size. What about the others? In lines 153-155 you have mentioned other important features? Why are they mentioned if they are not considered for the cities under the study? Such a list of indicators without the data creates a confusion: it is unclear which exact meanings of the indicators let to identify city as the medium-sized?

According to your instructions, characteristics and variables to be considered have been specified. The size of the population has been used in this paper debt to this is less dynamic variable and easier to measure for the approximation of this medium-sized cities. In the SEM analysis static variables are considered at moments in time, so the “growth of the population” is considered a dynamic variable, which would make measurement of this analysis difficult. Regarding territorial capacity and development plans, there is no homogeneous variable that allows measurement as it is a variable with an important qualitative component. Finally, the degree of specialization and industrialization is not available and there is no homogeneous variable that allows us to measure the differences between medium-sized cities.

Regarding other indicators mentioned, it is considered as potential to determine the size of a city, since it is recognized a city with companies with export potential, or the creation of new companies in a territory, are interesting when measuring them, But given the dynamism of these variables, it is not possible to compare these periods of time, so it is mentioned as other interesting variables, but they are not carried out in the analysis.

Thus, the indicators have been established from official and contrasted organizations such as the INE (Instituto Nacional de Estadística), the Institute of Statistics and Cartography of Andalusia and other organizations which have reliable data for an accurate measurement.

9. Even though the list of variables used to measure tourism development and socioeconomic development is impressive, it lacks scientific background. It remains unclear why these exact variables were chosen? How do I know that they represent for example tourism development component? Therefore, I recommend to strengthen the scientific background of the chosen indicators. Units of measurement of every indicator also should be mentioned.

The background for using this list of indicators comes from the previous research that can be found referenced:  Pulido-Fernandez, J.I., and Parrilla-Gonzalez J.A.. Determines the economic dynamism of tourism of a territory its socioeconomic development? An analysis through structural equation modeling. 2016. Revista de Estudios Regionales, 107, 87-120.

Before this paper, there was no clarity on what is considered Socioeconomic Development and what is considered Tourism Development, for this reason, the approach to the study is developed with these indicators from very reliable sources. This is one of the weaknesses of the analysis used as you have detected, but it has made it possible to fulfill the object of the research by measuring a direct relationship between tourism development and socioeconomic development.

Regarding the units of measurement, there are numerical values associated with quantities, in order to use the RRC. That is the value that the measurement allows.

10. What is the logic of global index? Does it have any scientific background?

To create the Global Index, the sum of the two indicators has been considered: tourism development and socioeconomic development. By showing a direct relationship between these indicators, a global index has been considered. In this way, a third more complete ranking has been obtained that yields interesting results linked to the development of medium-sized cities around the coast and large provincial capitals, thus promoting the hypothesis that tourism is a tool for socio-economic development establishing a key ranking for this analysis.

In the discussion section I have proceeded to clarify this term for greater clarity following your valuable recommendations.

11. Discussion section is missing.

Improvements have been incorporated following your valuable instructions

12. Conclusions present mainly technical results. It is recommended to specify how does the research contribute to the development of tourism of medium-sized cities. What do the results mean for the socioeconomic situation in these cities?

Improvements have been incorporated following your valuable instructions

13. Under the conditions of COVID-19 the tourism in Spain is the segment that has suffered painfully since March 2020. I clearly understand that the period of your research does not cover 2020, however in introduction and in discussion part it is valuable to provide your insights about the current context. In fact, the current situation adds extra relevance to your research.

Improvements have been incorporated following your valuable instructions

I hope the improved version can fulfill your instructions. Thank you very much for considering this manuscript in your review.

Sincerely,

Round 2

Reviewer 2 Report

The research is interesting. At the same time, the research and the model seem to be improved, but it is not enough. The crucial point is the interpretation of the chi-square statistic used to perform a perfect model fit test, which does not convince the final reliability of the results. The authors correctly tell that there could be an effect due to the sample size but this statement and not completely convincing with absent methodological and statistical references on this point. Could the authors provide some relevant references to support their point of view about the results? 

Author Response

Dear Reviewer,

Thank you very much for your comments. I hope this final version could be approved and published in this journal.

Sincerely,

Reviewer 3 Report

Thank you for the revised version of the manuscript. It looks much better now. 

Author Response

Dear Reviewer,

Thank you very much for your comments. I hope this final version could be approved and publish in this journal.

Sincerely,